# A Class 4-like Chromosomal Integron Found in *Aeromonas* sp. Genomospecies *paramedia* Isolated from Human Feces

**DOI:** 10.3390/microorganisms11102548

**Published:** 2023-10-13

**Authors:** Jesús Baltazar-Cruz, Rogelio Rojas-Rios, Violeta Larios-Serrato, Itza Mendoza-Sanchez, Everardo Curiel-Quesada, Abigail Pérez-Valdespino

**Affiliations:** 1Department of Biochemistry, Escuela Nacional de Ciencias Biológicas del Instituto Politécnico Nacional, Prolongación de Carpio y Plan de Ayala S/N, Col. Santo Tomás, Mexico City 11340, Mexico; jbaltazarc2@gmail.com (J.B.-C.); rogerteclab@gmail.com (R.R.-R.); viosdatafactory@gmail.com (V.L.-S.); 2Department of Environmental & Occupational Health, Texas A&M University School of Public Health, College Station, TX 77843, USA; itzamendoza@tamu.edu

**Keywords:** *Aeromonas* sp. genomospecies *paramedia*, clinical isolate, new integron, class 4-like integrase

## Abstract

Integrons are genetic elements that store, express and exchange gene cassettes. These elements are characterized by containing a gene that codes for an integrase (*int*I), a cassette integration site (*att*I) and a variable region holding the cassettes. Using bioinformatics and molecular biology methods, a functional integron found in *Aeromonas* sp. 3925, a strain isolated from diarrheal stools, is described. To confirm the integron class, a phylogenetic analysis with amino acid sequences was conducted. The integrase was associated to class 4 integrases; however, it is clearly different from them. Thus, we classified the associated element as a class 4-like integron. We found that the integrase activity is not under the control of the SOS or catabolic repression, since the expression was not increased in the presence of mitomycin or arabinose. The class-4-like integron is located on the chromosome and contains two well-defined gene cassettes: *aad*A1 that confers resistance to streptomycin and *lpt* coding for a lipoprotein. It also includes eight Open Reading frames (ORFs) with unknown functions. The strain was characterized through a Multilocus Phylogenetic Analyses (MLPA) of the *gyr*B, *gyr*A, *rpo*D, *rec*A, *dna*J and *dna*X genes. The phylogenetic results grouped it into a different clade from the species already reported, making it impossible to assign a species. We resorted to undertaking complete genome sequencing and a phylogenomic analysis. *Aeromonas* sp. 3925 is related to *A. media* and *A. rivipollensis* clusters, but it is clearly different from these species. In silico DNA-DNA hybridization (*is*DDH) and Average Nucleotide Identity (ANI) analyses suggested that this isolate belongs to the genomospecies *paramedia*. This paper describes the first class 4-like integron in *Aeromonas* and contributes to the establishment of genomospecies *paramedia.*

## 1. Introduction

Bacteria of the genus *Aeromonas* are Gram-negative bacilli distributed in different ecosystems, mainly in aquatic environments. *Aeromonas* have been isolated from drinking, bottled, irrigation and sea water, sewage, sediment, soil, food, animals and humans. *Aeromonas* species’ adaptability is a reflection of the plasticity of their genomes [1,2]. Genus *Aeromonas* includes 36 species and given the taxonomic complexity of the genus, different schemes of identification have been used for the assignment of species. Schemes include phenotypic tests, DNA–DNA hybridization, 16S rRNA RFLP and sequencing of the 16S rRNA gene; however, these methods are no longer accurate [3,4]. In some studies, the phylogenetic analysis is based on the amplification of a single gene [5,6]; however, this method is not accurate enough for species identification, particularly when the genus shows a high heterogeneity resulting from recombination and horizontal gene transfer [7]. At the present time, species assignment is based on the Multilocus Phylogenetic Analyses (MLPA) of housekeeping genes [8,9]. Recently, de Melo et al. [10] suggested high-resolution genome-wide analysis as an essential method for the unambiguous identification of *Aeromonas* isolates.

Many *Aeromonas* strains contain mobile genetic elements such as plasmids, insertion sequences, phages and integrons carrying resistance and/or virulence determinants. Some of these elements harbor open reading frames (ORFs) with unknown functions [11]. Integrons are stable genetic platforms for gene capture, storage and expression. These elements contain a conserved region composed of the *int*I gene encoding a site-specific recombinase, an *att*I recombination site and a promoter(s) (Pc) region. Next to this conserved portion, there is a variable region that contains cassette genes lacking promoters whose expression depends on their proximity to Pc [12]. Cassettes are flanked by *att*C sites required for their excision and subsequent insertion at the *att*I site [13].

Integrons are classified as class 1 to class 5 based on the sequences of their associated integrases; however, multiple unclassified integrases have been described [14,15]. When integrons are associated with mobile genetic elements that guarantee their dissemination, such as transposons or plasmids, they are regarded as mobile integrons (MIs). If elements are sedentary, they are chromosomal integrons (CI). Integrons that contain a large number of cassettes in their variable region are called superintegrons [16].

Class 1 integrons are described as the most prevalent and the most frequently linked to antibiotic resistance in Gram-negative bacteria [17], and class 2 integrons are second in abundance, while class 3 integrons have the lowest prevalence of all [16]. On the other hand, class 4 integrons are more complex structures having a large number of cassettes, which are normally not associated with antimicrobial resistance and have been described mainly in *Vibrio* [18]. Many reports indicate the presence of class 1 and 2 integrons in *Aeromonas* species [19,20,21], class 1 being more abundant than class 2 integrons. Nevertheless, the detection of class 2 integrons is difficult, and their prevalence has probably been underestimated [22]. Only one report of a class 3 integron in one strain of *A. allosacharophila* exists [23]. The current work made use of high-resolution genome analysis to describe a new *Aeromonas* genomospecies *paramedia* isolate. Unexpectedly, genome sequencing revealed the presence of a putative class 4 integrase and several cassettes associated. A complete characterization of this class 4-like integron was made.

## 2. Materials and Methods

### 2.1. Strains Isolation and Characterization

*Aeromonas* sp. 3925 was collected from a stool sample of a patient with gastroenteritis who attended the Public Health Service at Hidalgo state, Mexico. The sample was provided by the patient in order to confirm the diagnostic and was not obtained directly from the patient himself. Initially, the isolated was classified as *A. media* by bacteriological tests, *gcat* amplification and 16S rRNA PCR-RFLP [24]. The isolate was ampicillin resistant and able to grow in mineral medium. The strain was preserved in Luria (DIBICO, Cuautitlán, Edomex, Mexico) broth supplemented with 20% glycerol and stored at −70 °C.

### 2.2. Species Identification Based on Conventional Multilocus Phylogenetic Analyses (MLPA)

Genomic DNA of the bacterial isolate was extracted from Luria broth overnight cultures using the protocol described by Green and Sambrook [25]. For the PCR amplification of the *gyr*B, *gyr*A, *rpo*D, *rec*A, *dna*J and *dna*X genes, primers and amplification conditions reported by Martinez-Murcia et al. [8] were used (Appendix A). Amplicons were sequenced by the Sanger method using the ABI3730XL system. Sequences were edited for quality check and trimming using SnapGene Software v. 3.2.1. The resulting dataset together with another 36 sequences of the same *Aeromonas* genes obtained from the GenBank (Appendix A) were aligned and concatenated using Seaview V.5.0.5 [26]. *Oceanimonas* spp. GK11 was used as an out-group. Gene sequences were used for the phylogenetic analysis, which was performed using the Maximum Likelihood (ML) method, as implemented in the program PHyML 3.0. ML bootstrap supports were calculated after 100 iterations. The nucleotide substitution evolution model was automatically selected by Smart Model Selection (SMS) using AIC [27]. The resulting phylogenetic tree was displayed using FigTree v. 1.4.4.

### 2.3. Genome Sequencing and MLPA

Genomic DNA was extracted, and concentration and purity were determined by NanoDrop^TM^ 2000 (Thermo Fisher Scientific, Waltham, MA, USA) and Qubit^R^ 2.0 (Life Technologies, Carlsbad, CA, USA). DNA was sequenced using Illumina-compatible adapters with unique barcodes ligated onto each sample during library construction. Sequencing libraries were prepared using the NEBNext Ultra DNA Library Prep Kit (NEB, Ipswich, MA, USA) and subsequently were pooled in equimolar concentrations for multiplexed sequencing on the Illumina MiSeq platform with 2 × 150 run parameters. Sequencing run data were analyzed and demultiplexed. Sequences quality was assessed using FastQC v.0.11.8 [28]. The raw paired readings were trimmed and filtered with the Trimmomatic program v.0.39 [29]. Reads were de novo assembled using Spades v.3.13.0 [30], and a quality control assembly test was conducted with QUAST V.5.2 [31]. Contigs were organized using Mauve software v.2.4.0 [32]. Coding sequences were predicted using Prokka V.1.14.6 [33] and Rapid Annotation using Subsystem Technology (RAST) V.2.0 [34]. Plasmids in the genomic data were identified and assembled using plasmidSPAdes [35]. The genes of interest were visualized using Artemis V.17.0.1 [36].

The sequences of *atp*D, *dna*J, *dna*K, *dna*X, *glt*A, *gro*L, *gyr*A*, gyr*B, *met*G, *pps*A, *rad*A, *rec*A, *rpo*B, *rpo*D and *tsf* housekeeping genes were used for MLPA analysis. Alignment was completed with the above-mentioned 36 references sequences plus 2 sequences of putative *Aeromonas* sp. genomospecies *paramedia,* 2 sequences of *A. rivipollensis* and 1 sequence of *A. media* to a total of 41 sequences with *Oceanimonas* spp. GK11 as an external group, using the Muscle algorithm with SeaView program V.5.0.5. The concatenated genes alignment resulted in a 27,276 bp sequence that was used for phylogenetic analysis with the ML method. Execution of the analysis was completed with the online program PhyML. The robustness of the relationships was obtained using a transfer bootstrap of 100 iterations [37]. A nucleotide substitution evolution model was completed as mentioned above. Visualization and editing of the phylograms was carried out with the FigTree program V.1.4.4.

### 2.4. Phylogenomic Analysis by isDDH and ANI

The genomes of *Aeromonas* strains were compared by in silico DNA–DNA hybridization (*is*DDH) and average nucleotide identity (ANI) methods to establish phylogenetic relationships. The ANI of the genomes was calculated with the online program ANI calculator (Rodriguez, Kostas Lab, Riverside, CA, USA) with default parameters, and the analyzed genomes were considered to belong to the same bacterial species with an ANI value ≥ 95%, as previously established [38,39]. In silico DDH values were calculated with the GGDC 2.1 program [40]. Two available genomes, *A. media* 4234 and *A. rivipollensis* KN-Mc-11N1, closely related to the subject species were used. Genomes showing *is*DDH similarity values ≥ 70% were considered to be members of the same species, as previously recommended.

### 2.5. Genome Analysis and Comparative Annotations

The subsystems annotated for RAST were examined to establish their functional roles. Subsequently, sequences were subjected to BLASTN (V.2.14.1) analysis (with a minimum threshold of 80% nucleotide identity) to determine sequence similarity and database matches for DNA sequences [41]. ORFs were confirmed by a BLASTx search in the NCBI non-redundant protein database [34]. To continue with the analysis, the Conserved Domains platform (NCBI) was used. TAfinder 2.0, ResFinder 3.2 and Plasmid Finder were used to predict toxin and antitoxin genes, resistance markers and replicons, respectively [42,43,44]. Integrons and *att*C sequences were searched using Integron Finder 2.0.2 [45] and HattCI [46]. Promoter analysis was completed with BPROM (Softberry, Mt Kisco, NY, USA) [47] and promoter hunter (phiSITE) [48] tools. Visualization of the genome was completed with Proksee [49].

### 2.6. Characterization and Functionality of the Integron

A mating experiment using *Escherichia coli* S17-1 λ *pir* RP4 (pICV8) as a donor and *Aeromonas* sp. 3925 as a recipient was performed to assess integrase functionality. Transconjugants were selected on nutritive plates containing 50 μg/mL ampicillin plus 75 μg/mL zeocin. Transformant colonies from a plate were pooled and extracted for plasmids. Pooled plasmids were used in turn to transform *E. coli* DH5α (Invitrogene, Carlsbad, CA, USA) to streptomycin resistance in order to assess the mobility of the cassette *aad*A1 from a class 4 chromosomal integron to pICV8 [50,51].

To evaluate the Pc activity, streptomycin minimal inhibitory concentration (MIC) was determined by the microdilution method, following the protocol of the Clinical and Laboratory Standards Institute [52]. *A. hydrophila* 6479 and *E. coli* C3297 bearing an *aad*A2 cassette were used as resistant controls [22,50]. *E. coli* DH5α was used as the sensitive control strain. 

The integrase gene and the three first cassettes of the class 4 integron (*intI-orf*1*-orf*2*/lpt-aad*A1) were amplified (Appendix A) and cloned into the pBBR1MCS-3 vector [53]. Briefly, 5 μL of Gibson Assembly Mix (New England BioLabs. Beverly, MA, USA) was mixed with 0.2 pmoles of fragment and *SmaI* linearized pBBR1MCS-3 to a final volume of 10 μL. Reaction was incubated at 50 °C for one hour. After incubation, 0.5 μL of the assembly mix was used to transform *E. coli* DH5α−competent cells. Transformants were selected on Luria agar plates containing tetracycline (20 μg/mL). Plasmids from transformants were isolated and digested to corroborate the presence of the pBBR1MCS-3::*intI-orf*1*-orf*2*/lpt-aad*A1. The functionality of the *aad*A1 cassette was confirmed by the appearance of streptomycin-resistant colonies after transformation.

### 2.7. Analysis of intI4-like Integrase Expression by RT-PCR

*int*I4-like integrase gene expression levels were evaluated through real-time RT-PCR from cultures grown under five different conditions: (a) in the presence of 256 μg/mL streptomycin; (b) 2 μg/mL mitomycin; (c) 1% arabinose; (d) senescent culture; or (e) in the absence of any stimulus. Cultures were stimulated by the addition of antibiotic or sugar in the early logarithmic growth phase. Total RNA from cultured cells was extracted using the SV total RNA isolation system (Promega, Madison, WI, USA). Purity and concentration were evaluated by spectrophotometry (Nanodrop 2000 Thermo-Scientific). RNA integrity was assessed by electrophoresis in agarose gels supplemented with 0.5% sodium hypochlorite. RNA was reverse transcribed using the cDNA Synthesis & Go Kit (MPI, Dresden, Germany) and amplified using the LightCycler 480 SYBR Green I Master (Roche, Mannheim, Germany). Each experimental condition was repeated three times in assays performed in triplicate. Expression values for *int*I4-like and *rec*A under each stimulus were normalized to the *gcat* gene, and relative expression levels were calculated using the 2^−ΔΔC^T method [54]. The *rec*A gene, known to be induced by SOS response, was used as a control [55,56]. Primers used in real-time RT-PCR assays are shown in Appendix A. Two-way ANOVA, followed by Dunnett’s test, was used to compare the relative gene expression values for *rec*A and *int*I4-like (**** *p* < 0.0001).

## 3. Results

### 3.1. Species Assignment of Aeromonas sp. 3925

*Aeromonas* sp. 3925 was previously identified by 16S-rDNA PCR-RFLP-like *A. media* [24]; nevertheless, this identification scheme has generated misassignments to the species level. Therefore, re-identification was carried out by concatenated MLPA analysis with six genes, but still the species could not be defined. Accordingly, the number of housekeeping genes was increased to fifteen in the MLPA (*atp*D, *dna*J, *dna*K, *dna*X, *glt*A, *gro*L, *gyr*A, *gyr*B, *met*G, *pps*A, *rad*A, *rec*A, *rpo*B, *rpo*D, *tsf*). This analysis revealed that *Aeromonas* sp. 3925 was closely related to *A. media* and *A. rivipollensis* with high bootstrap values for both species (Figure 1). Phylogenetic relationships with these species were defined, increasing the number of reference sequences to resolve the tree topology. Results showed the grouping of strain 3925 in the clade including *A. media* UTS15 and *A. media* 1086C strains with a support value of 100 at the node. This phylogenetic configuration separates strain 3925 from the representative clades of *A. rivipollensis* and *A. media*-type strains and places it in a different clade. Finally, the genomes of *A. rivipollensis* (KN-Mc-11N1) and *A. media* (CECT 4234) were compared with *Aeromonas* sp. 3925 by ANI and *is*DDH, resulting in DDH values of 54% and 58%, as well as ANI values of 93.8% and 94.5%, for *A. media* and *A. rivipollensis*, respectively. These similarities are clearly under the threshold values proposed for species delimitation (95–96% for ANI and 70% for DDH). Consequently, these results contribute to the description of the new species *Aeromonas* sp. genomospecies *paramedia* proposed by Talagrand-Reboul et al. [57].

### 3.2. Genome Analysis

*Aeromonas* sp. genomospecies *paramedia* 3925 contains a genome of 4.78 Mb with an observed GC content of 61.5%. It also carries two plasmids p*Aer*XVI (33.7 Kb) and p*Aer*XVII (22.2 Kb) (Figure 2). Annotation analysis revealed 4436 coding genes. The whole-genome draft has been deposited at GenBank under accession number NZ_JAAROG000000000.1 (Genome assembly ASM2309351). Plasmids were named p*Aer*, which denotes their provenance, followed by roman numerals that refer the consecutive order of the plasmids described in our collection of *Aeromonas* strains [58].

### 3.3. Plasmid Analysis

p*Aer*XVII contains *tra*M, L (*vir*D4), K, J, H, I, D (*vir*B4) and C genes associated to mobility by conjugal transfer; however, the conjugation gene machinery is incomplete. This plasmid has a toxin–antitoxin system (*hin*A-*hin*B) and a gene encoding a class VII phage endonuclease. In this element, nineteen ORFs code for hypothetical proteins.

p*Aer*XVI carries the *trb* operon *(trbB*, *D*, *E*, *J*) that encodes most of the apparatus for mating pair formation and other genes associated to plasmid mobilization. Two phage proteins, a serine protease and sixteen ORFs, are encoded by this plasmid. It is important to highlight that genes contained in both plasmids could complement each other for their conjugal transfer, since both plasmids bear an *ori*T. This would be a very special case of plasmid transfer with two mobilizable plasmids. No antibiotic resistance genes were found in these plasmids.

### 3.4. Integron Characterization

The genome analysis by RAST suggested the presence of a chromosomal class 4 integron integrase. To confirm the integrase class, a phylogenetic analysis with amino acid sequences of these enzymes from different γ-proteobacteria was carried out. The taxonomic distribution of integrases is very heterogeneous, and their wide diversity has a big influence in the classification. Initially, the integrase was associated to class 4; however, it is clearly in a separate clade (Figure 3). In a second analysis, sequences from bacteria recovered from Antarctic and marine environments were included. This new analysis confirms a close relationship of this integrase with class 4 integrases (Appendix A).

The molecular size of the chromosomal integron of *Aeromonas* sp. 3925 is 8.3 Kb; this element contains ten ORF and eight *att*C recombination sites. The *att*C sites display variable (60–99 bp) length (Figure 4). Two cassettes encode known proteins; *aad*A1 codes for streptomycin resistance and *lpt* gene codes for a lipoprotein. The rest of the cassettes encode hypothetical proteins; however, some of these proteins contain recognized functional domains. The *orf*6 encodes a protein with a nuclease GIY-YIG domain, *orf*8 codes for a membrane protein, the protein encoded by *orf*9 includes an endonuclease domain, and finally, *orf*10 codes for a cold shock domain-containing protein. Two cassettes seem to be translationally coupled (*orf*7–*orf*8), and between *orf*2 and *orf*3, there is no *att*C. This cassette arrangement has not been described before.

The regulatory region of the integron has the following characteristics: (i) a primary recombination site *att*I (227 bp) with a core sequence GTTRRRY was found between the conserved and variable regions; (ii) P*int*I is located 27 bp upstream the start codon of the *int*I gene; (iii) four putative promoter cassettes (Pc 4L-1 to 4L-4) with polymorphisms in the canonical sequence of previously described promoters were identified (Figure 4; Appendix A); (iv) two putative integrase regulating sequences were identified; *lex*A and CRP boxes were associated with the SOS response and carbon catabolite repression system, respectively. 

In order to explore the possible expression of integrase and the gene cassettes, the ribosomal binding sites (RBSs) or Shine–Dalgarno sequences (SDs) in the integrase and the gene cassettes were analyzed (Appendix A).

### 3.5. Cassette Expression 

A cassette’s activity depends on upstream active promoter sequences. Since the *Aeromonas* genomospecies *paramedia* 3925 genome lacks other genes conferring streptomycin resistance, expression of the *aad*A1 cassette was evaluated. *Aeromonas* sp. 3925 displayed an MIC of 64 μg/mL streptomycin. *aad*A1 expression was confirmed after transforming *E. coli* DH5α pBBR1MCS-3::*intI-orf*1*-orf*2*/lpt-aad*A1. The transformants were resistant to 128 μg/mL of the antibiotic, which confirms the functionality of the Pc promoter.

### 3.6. Expression of Class 4-like Integrase 

To evaluate the cassettes’ mobility, the pICV8 plasmid was introduced to *Aeromonas* sp. genomospecies *paramedia* 3925. If the integrase was expressing, excision of the *aad*A1 cassette and its integration into the *att*I1 site of pICV8 could occur. To test this, plasmids from the transconjugants were extracted and used to transform in *E. coli* DH5α to streptomycin resistance; however, transformants carrying pICV8::*aad*A1 were not recovered. This may be due to a lack of integrase expression or to the inability of class 4 integrase to recognize the *att*I1 region. As revealed by RT-PCR, the gen *int*I4-like expresses constitutively in bacteria grown in the absence of stimulus. Overexpression in the presence of streptomycin, mitomycin or arabinose was not observed; senescent cultures showed a very slight increase in the expression (Figure 5).

## 4. Discussion

Taxonomy of the genus *Aeromonas* is complex and has experienced frequent changes over time [1]. Technological advancements have made it possible to perform consistent species identification. The massive sequencing era makes it now possible to achieve a deeper and more robust taxonomy [59,60]. The use of whole genome information has revealed the limitations of techniques like MLPA. In this work a re-identification of *A. media* 3925 was performed. Results of several phylogenomic analyses (ANI, *is*DDH and PhyML) supported the conclusive identification of the strain [38,61]. Even though results showed a close phylogenetic relationship of *Aeromonas* sp. 3925 with *A. media* and *A. rivipollensis*, it could not be conclusively assigned to any of these species. In order to clarify this issue, it was necessary to increase the number of *A. media* and *A. rivipollensis* representative sequences in the analyses. This allowed us to group strain 3925 together with *A. media* UTS15 and *A. media* 1086C in a new species termed *Aeromonas* sp. genomospecies *paramedia* [57]. It is worth emphasizing the need for high-resolution genome analysis for the identification of ambiguous *Aeromonas* strains as well as the inclusion of as many genomes as possible in order to support the genetic identification [10].

Genome analysis showed the existence of a gene homologous to the one coding for class 4 integrase; however, phylogenetic analysis revealed that some evolutionary changes occurred, placing this integrase in an independent clade. Therefore, we designated this gene, together with their adjacent cassettes arrangement, a class 4-like integron. Antelo et al. [15] performed a metagenomic analysis of integron-associated integrases in Antarctic bacteria, which led to the discovery of a wide variety of these site-specific recombinases. The integrase described in this work clustered in group II like *Vibrio’s* class 4 integrases. The great abundance of integrons observed in environmental and clinical isolates reveals new integron classes [62].

The report of this integron is important for two reasons. First, the integron diversity in *Aeromonas* is limited, since only classes 1, 2 and 3 [20,22,63] have been described. Secondly, the variable region is not only linked to antimicrobial resistance. Abella et al. [14] reported a wide variety of cassettes in marine bacteria, which suggests that integrons may be associated to bacterial adaptation phenomena independent from antibiotic resistance. 

Class 4-like integron encloses a gene cassette coding for a protein belonging to the Lpt family (pfam07553), which are phage-encoded proteins responsible for the delivery of phage DNA to the target cell. The *aad*A1 gene encodes an adenylyltransferase that confers streptomycin resistance. Variants of *add*A cassettes are very common in integrons [64]. Other cassettes encoding hypothetical proteins include specific domains [65]. The *orf*6, for example, encodes for a protein with a GIY-YIG nuclease domain. These types of nucleases are engaged in DNA recombination and repair [66]. The *orf*9 encodes a protein with an endonuclease domain (Endonuclease_DUF559), belonging to the nuclease superfamily. This family, as stated in the NCBI, “includes a very short patch repair (Vsr) endonucleases, archaeal Holliday junction resolvases, MutH methyl-directed DNA mismatch-repair endonucleases and catalytic domains of many restriction endonucleases”. The protein encoded by *orf*10 includes a “cold shock domain”. This domain is found in small proteins with affinity to single-strand DNA, which are stress-responsive proteins induced by a sudden decrease in temperature. These proteins work also under normal conditions, playing other biological roles [67,68]. 

Different reports document variability in the expression levels of the integron cassettes. This variability results from polymorphisms at the promoter sequences as well as the distances between the individual cassettes and the Pc promoter [69]. Four putative cassette promoters (4L-1 to 4L-4) responsible for cassette expression were identified upstream from the cassettes. These promoters share some characteristics with those described in other integron classes. Pc and P2 are found in class 1 integrons. Pc is located in class 1 integrons inside the integrase-coding region, 223 to 252 bp from the GTT site at the *att*I1 region, while P2 is 119 bp downstream Pc [70]. A promoter Pc 4L-1 shared the location of the class 1 Pc promoter, 264 bp away from the integration triplet GTT; however, its -10 and -35 consensus sequences show nucleotide changes compared to the described variants [71]. At 37 bp downstream, between the integrase gene and the *att*I, there is a second promoter Pc 4L-2; its position and sequence contrast with other class 1 P2 promoters. The class 2 promoters regions contain at least five promoters [72,73], four of which are embedded in the *att*I2 region; however, the sequences of these promoters are different from those of the promoters at the *att*I described in this work. Chromosomal integrons in *Vibrio* sp. contain only one Pc promoter, which is located 65 bp upstream from the recombination site *att*IA. *Vibrio*’s Pc promoter is activated by the global regulator cAMP-CRP, which binds directly with a CRP box located 41 bp upstream at the transcription start site of the cassettes, overlapping with the -35 promoter region [74] much like how Pc and 4L-4 overlap with a putative CRP box. We confirmed the expression of the *aad*A1 gene, found at the fourth position of the integron, which indicates that at least one Pc 4L, the promoter is functional. Cassettes are mobilized by an integrase that recognizes recombination sites to integrate (*att*I × *att*C) or excise (*att*C × *att*C) them from the variable region. The integrase is expressed, but we could not demonstrate its competence to insert the *aad*A1 cassette to the *att*I site of pICV8. This could be due to the inability of the enzyme to recognize class 1 *att*I sites or to a failure in integrase subexpression. A detailed analysis of the region upstream the integrase gene revealed the absence of direct repeats (DR1 and DR2) or inverted repeats (R and L). Only the excision site (GTT) was found. The integrase requires a core sequence of six to eight nucleotides, but only the GTT triplet is absolutely conserved [13,75]. 

Class 1 integrase expression is regulated by the SOS response. This is a global reaction to DNA injury in which the cell cycle is arrested and DNA repair genes are expressed. The master regulator of SOS response is the LexA repressor [55,76]. Bioinformatic analysis revealed a *lex*A box next to the -10 region of the P*int*I4-like promoter; however, overexpression of the *int*I4-like gene was not observed in bacteria grown in the presence of streptomycin or mitomycin. Our results indicate that the recombinase activity is not under control of the SOS response [50]. A CRP box associated to catabolite repression was detected, but it could not detect any changes in *int*I4-like expression in cells grown in the presence of arabinose or glucose, which indicates that integrase expression is not under catabolic repression [77]. Additional work will be necessary to extend our understanding of the mechanisms controlling the dynamics of class 4-like integron. 

## Figures and Tables

**Figure 1 microorganisms-11-02548-f001:**
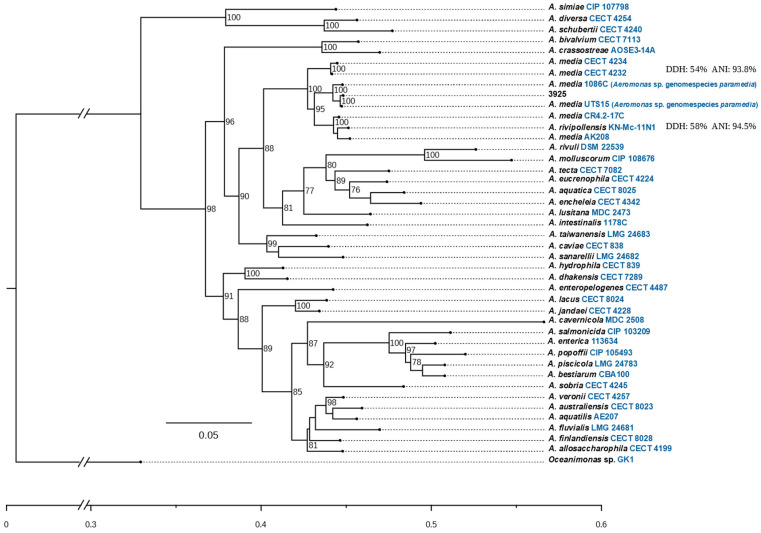
Phylogenetic tree reconstructed from the alignment of 15 housekeeping genes (*atp*D, *dna*J, *dna*K, *dna*X, *glt*A, *gro*L, *gyr*A, *gyr*B, *met*G, *pps*A, *rad*A, *rec*A, *rpo*B, *rpo*D, *tsf*). Numbers next to nodes indicate bootstrap values (100 replicates). ANI and DDH represent genetic similarity between genomes of *Aeromonas* sp. 3925, A*. rivipollensis* and *A. media*.

**Figure 2 microorganisms-11-02548-f002:**
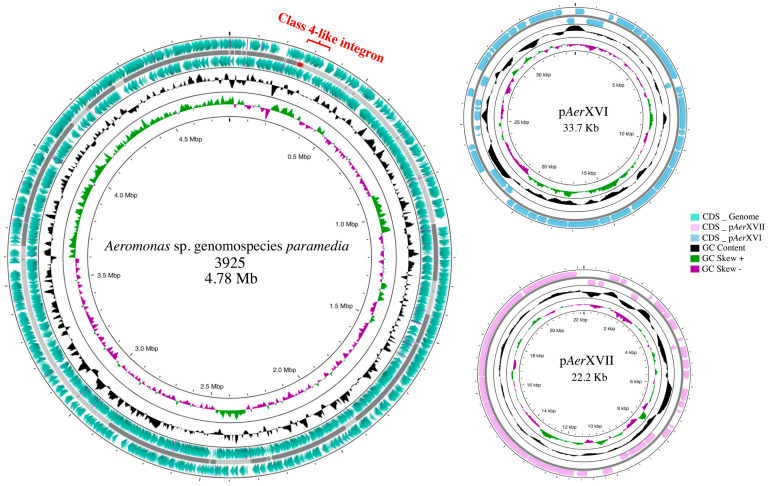
Schematic circular representation of complete genome and plasmid sequences of *Aeromonas* sp. genomospecie*s paramedia* 3925. The raw paired readings were trimmed and filtered with the Trimmomatic program v.0.39. Reads were de novo assembled using Spades v.3.13.0, and a quality control assembly test was conducted with QUAST V.5.2. Contigs were organized using Mauve Version 2.4.0 software. The coding sequences were predicted using RAST V 2.0. Visualization of the genome was completed with Proksee.

**Figure 3 microorganisms-11-02548-f003:**
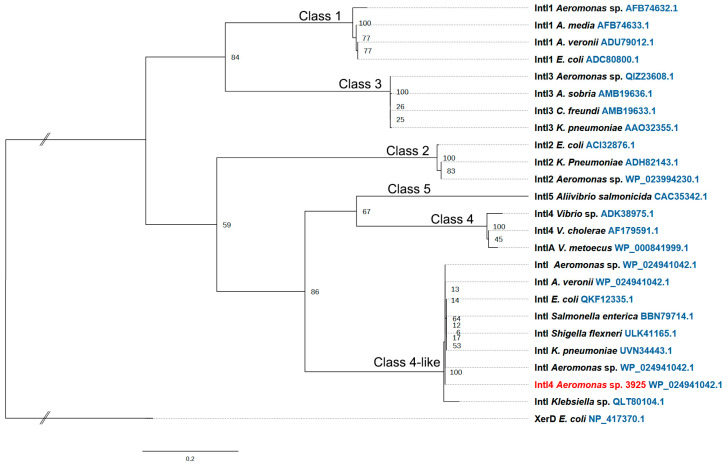
Phylogenetic relationship of the integron IntI in the γ-proteobacteria. This tree was made using amino acid sequences of integron integrases. Bootstrap support values represent the consensus of distance neighbor-joining trees obtained from 100 replicates of the data set. Branch lengths are proportional to the number of evolutionary changes that have occurred based on genetic distance; the scale bar indicates the number of substitutions per site. Reference sequences were included in the analysis (IntI1, IntI2, IntI3, IntI4, IntI5, and XerD sequences).

**Figure 4 microorganisms-11-02548-f004:**
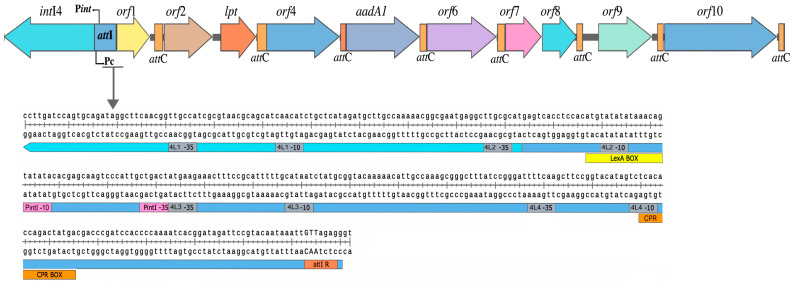
Schematic representation of class 4-like integron from *Aeromonas* sp. genomospecies *paramedia* 3925. Integron structure comprises the gene *int*I4-like that codes for integrase followed by the primary recombination site *att*I and the variable region conformed by eight cassettes. P*int*I and Pc putative promoters (4L-1 to 4L-4) are shown in the sequence of integrase and *att*I region.

**Figure 5 microorganisms-11-02548-f005:**
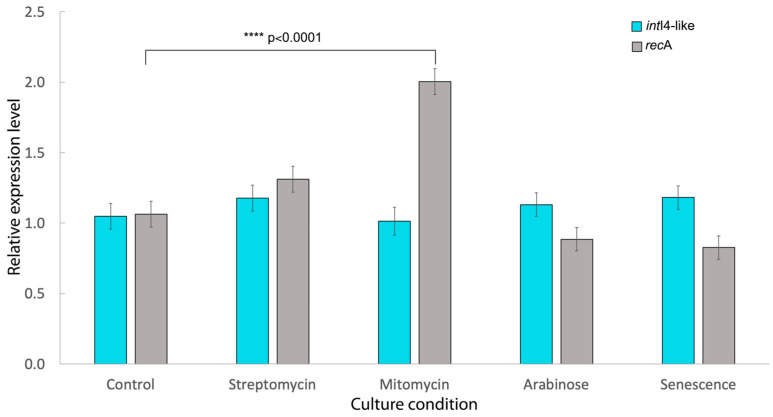
Expression analysis of *int*I4-like gene. qRT-PCR was used to measure the transcript levels of the *int*I4-like gene under different conditions. Expression levels were normalized to the housekeeping gene *gcat*. *rec*A was used as a control in the induction of SOS response. Standard deviations from three independent repeated trials are indicated by the error bars. Differences were determined using two-way ANOVA followed by Dunnett’s test (**** *p* < 0.0001).

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
