# Peer review of "A Class 4-like Chromosomal Integron Found in Aeromonas sp. Genomospecies paramedia Isolated from Human Feces"

_microorganisms, 2023, doi:10.3390/microorganisms11102548_

Round 1

Reviewer 1 Report

Please revise your paper in response to the minor comments in the attached file.

Too little English language editing is required, as noted in the attached file.

Author Response

Dear reviewer, thanks for your observations and comments.

As suggested abstract (lines 13 to 30 )was modified to make it more structured and modifications in the text were included (lines 387 to 390 and 403).

Thank you

Reviewer 2 Report

The study proposed the identification of a new category of class 4-like integron from Aeromonas sp. genomospecies paramedia 3925. I have the following concerns for the authors

1-The author should locate key annotations in Figure 2 not only showing a representation of sequenced genome

2- Have you performed any phenotypic analysis of the isolated strain of Aeromonas, I would like to see their antibiotic sensitivity to aminoglycosides to evaluate the functionality of this integron

3- Consider this reference for the description of integrons in your introduction https://doi.org/10.3390/antibiotics10050488

Moderate editing of English language required

Author Response

Dear reviewer thanks for your observations and comments.

  1. Figure 2 was modified as suggested. 
  2. Phenotypic characterization of antibiotic resistance is included in lines 301 to 306.
  3. The article by El-Baz, A.M et al., 2021 was included in the discussion (lines 367 to 357). Thank you

Reviewer 3 Report

Please see the attached file for the Reviewer's Report on the manuscript ID: microorganisms-2627378.

Author Response

Dear Reviewer

Thank you for the paper review.  In the comments and suggestions section it refers to  an "attached file for the Reviewer's Report" however I can´t see the link or file. Could you attached file? 

Thank you

Abigail

Reviewer 4 Report

The manuscript by J. Baltazar-Cruz et al. entitled as "A class 4-like chromosomal integron found in Aeromonas sp. genomospecies paramedia isolated from human feces." is devoted to the discovery of a new integron in Aeromonas sp. genomospecies paramedia. The manuscript is generally well written and seems interesting. Although there are no major issues. The minor issues are below:

L001, L031: Omit dot at the end of the title and keywords.

L028: Replace hybridization with identity for ANI.

L078: Add symptoms and diagnosis for the patient.

L083: Add the manufacturer of LB broth.

L085: What is the difference between multilocus phylogenetic analysis and multilocus sequence analysis? It seems that there are two different MLSAs: 1st based on 6 genes, and 2nd based on 15 genes.

L092: Where I can find a list of 36 sequences?

L093: Which Oceanimonas was used? Strain name, accession number?

L095: Typo - ML.

L098: Omit ref. 28.

L099: There is no MLST was done, only MLSA.

L101: Add Nanodrop 2000 and add Qubit model also.

L102-L103: Add names of libraries used.

L106: Add ref. for FastQC instead of Babraham Bioinformatics.

L115: Again, MLPA or MLSA?

L116: Which 41 reference sequences?

L117: Two different versions of SeaView have been used? 5.0.5 and 4.6 (L093)? Why is this?

L119 and L322: Typo - PhyML.

L124: Add what is isDDH and ANI here, not just in the abstract.

L129: Add ref, for GGDC 2.1.

L129: Shorten Aeromonas to A.

L142: Add references for Softberry, BPROM and promoter hunter.

L152: Add manufacturer for E. coli DH5alpha.

L163: Italicize SmaI.

L183: Upperscript for T

Figure 1: Change bootstrap values to percent, as it done in Figure 3.

L237: Omit '

The English could have been better.

Author Response

Dear reviewers, thanks for your observations and comments.

  1. Line 001, L31: Dot was omitted.
  2. Line 28 DDH and ANI, meaning of the abbreviation was removed.
  3. Diagnosis was gastroenteritis. This note is in the text now. However, we have not data concerning the patient´s symptoms.
  4. Line 83 Manufacturer was added.
  5. L085 (now line 87) The term MLST is often used, in general, for sequencing multiple housekeeping genes, whereby the analysis is not necessarily based on allele numbering but on the calculation of total sequence similarity. Furthermore, they proposed the use of the term multilocus sequencing analysis (MLSA) when strains are clustered based upon sequences. MLSA of representative strains of a genus offers the opportunity to incorporate concatenated multigene phylogenies into bacterial systematics, as already recommended in 2002 by the ad hoc committee for the re-evaluation of the species definition in bacteriology and by other authors. Since the DNA sequence data is subsequently subjected to phylogenetic analysis, the term multilocus phylogenetic analysis (MLPA) has recently been considered more appropriate (Martínez Murcia et al., 2011). Accordingly, we changed MLSA to MLSP. MLST was removed.
  6. L092 (now line 95). Data are included in supplementary material (Table S2) as suggested .
  7. L093 (now line 96). Name of the strain was added Oceanimonas GK1, GenBank access is the Table S2.
  8. Line 95 (97). ML done
  9. Line 98 (now line 100). Reference 28 was omitted
  10. Line 99 (now line 102) MLST was change by MLPA
  11. Qubit model was added
  12. Line 106-108. Name of the libraries was added. NEBNext Ultra DNA Library Prep Kit
  13. Line 110. Reference was added
  14. Line 115 (now line 119) MLST was change by MLPA
  15. Line 116 (now lines 120-122). Reference sequences were better specified
  16. Line 117 (now 123). SeaView version was modified
  17. Line 119 (now 123). PhyML done
  18. Line 124 (now 131). The significance of DDH and ANI were added and corrected.
  19. Line 129 (now136). Reference was added
  20. Line 129 (now 137) Aeromonas was shorten to A.
  21. Line 142 1(now 50-151) Reference was added
  22. Line 152 (now 161) Manufacturer of coli DH5a was added
  23. Line 172 SmaI was italize.
  24. Figure 1 was modified as suggested
  25. Line 237 apostrophe was removed

Changes were made in the article

Round 2

Reviewer 2 Report

I am pleased to accept the manuscript in its present form

Minor editing of English language required